# The Impact of Exposure to Iodine and Fluorine in Drinking Water on Thyroid Health and Intelligence in School-Age Children: A Cross-Sectional Investigation

**DOI:** 10.3390/nu16172913

**Published:** 2024-08-31

**Authors:** Siyu Liu, Xiaomeng Yu, Zhilei Xing, Peisen Ding, Yushan Cui, Hongliang Liu

**Affiliations:** 1Department of Epidemiology and Health Statistics, School of Public Health, Tianjin Medical University, 22 Qixiangtai Road, Heping District, Tianjin 300070, China; liusiyu@tmu.edu.cn (S.L.); yxm8988@tmu.edu.cn (X.Y.); zhilei1118@tmu.edu.cn (Z.X.); peisending@tmu.edu.cn (P.D.); 2Tianjin Institute of Medicine Science, 79 Duolun Road, Heping District, Tianjin 300020, China; 3Institute of Environment and Health, Tianjin Centers for Disease Control and Prevention, 6 Huayue Road, Hedong District, Tianjin 300011, China

**Keywords:** iodine, fluorine, drinking water, school-age children, thyroid, intelligence

## Abstract

Iodine and fluorine, as halogen elements, are often coexisting in water environments, with nearly 200 million people suffering from fluorosis globally, and, in 11 countries and territories, adolescents have iodine intakes higher than that required for the prevention of iodine deficiency disorders. It has been suggested that excess iodine and/or fluorine can affect thyroid health and intellectual development, especially in children, but their combined effect has been less studied in this population. This study investigated 399 school-age children in Tianjin, China, collected drinking water samples from areas where the school-age children lived, and grouped the respondents according to iodine and fluorine levels. Thyroid health was measured using thyroid hormone levels, thyroid volume, and the presence of thyroid nodules; intelligence quotient (IQ) was assessed using the Raven’s Progressive Matrices (CRT) test; and monoamine neurotransmitter levels were used to explore the potential relationship between thyroid health and intelligence. Multiple linear regression and restricted cubic spline (RCS) analyses showed that iodine and fluorine were positively correlated with thyroid volume and the incidence of thyroid nodules in school-age children, and negatively correlated with IQ; similar results were obtained in the secondary subgroups based on urinary iodine and urinary fluoride levels. Interaction analyses revealed a synergistic effect of iodine and fluorine. A pathway analysis showed that iodine and fluorine were negatively associated with the secretion of free triiodothyronine (FT3) and free tetraiodothyronine (FT4), which in turn were negatively associated with the secretion of thyroid-stimulating hormone (TSH). Iodine and fluorine may affect IQ in school-aged children through the above pathways that affect thyroid hormone secretion; of these, FT3 and TSH were negatively correlated with IQ, whereas FT4 was positively correlated with IQ. The relationship between thyroid hormones and monoamine neurotransmitters may involve the hypothalamic–pituitary–thyroid axis, with FT4 hormone concentrations positively correlating with dopamine (DA), norepinephrine (NE), and 5-hydroxytryptophan (5-HT) concentrations, and FT3 hormone concentrations positively correlating with DA concentrations. Monoamine neurotransmitters may play a mediating role in the effects of iodine and fluoride on intelligence in schoolchildren. However, this study has some limitations, as the data were derived from a cross-sectional study in Tianjin, China, and no attention was paid to the reciprocal effects of iodine and fluorine at different doses on thyroid health and intelligence in schoolchildren in other regions.

## 1. Introduction

As halogen elements, fluorine and iodine have similar chemical properties. Fluorine and iodine often coexist in the environment. In China, groundwater with a high iodine content (TI > 100 μg/L) is mainly distributed in the Huanghuaihai–Yangtze River Delta Plain Belt, the bedrock foothills in South China, and the Loess Plateau in Northwest China [1]. High-fluorine groundwater is widely distributed in Northern China, Northeastern China, and Northwestern China in 29 provinces, cities, and autonomous regions [2,3]. Iodine, as one of the essential trace elements, plays a vital role in human growth and development. Iodine regulates the development of children’s nervous system by affecting the secretion of thyroid hormones (THs) [4]. A small amount of fluorine intake helps reduce dental caries [5] but excessive intake will also affect the development of children’s nervous systems [6]. Previous studies have shown that nearly 200 million people worldwide suffer from fluorosis [5]. The iodine intake of adolescents in 11 countries and regions is higher than the required level for preventing iodine deficiency [7]. Therefore, excessive fluorine and/or iodine exposure poses a serious threat to the growth and development of human beings, especially adolescents [8].

Epidemiological studies have found that the combination of high fluorine and high iodine can synergistically cause damage to the body, and exposure to high doses of fluorine and iodine will have a certain impact on human thyroid health, such as the formation of goiters and thyroid nodules [8,9]. The main mechanism is that fluorine can reduce the concentration of iodine in thyroid follicular cells by disturbing thyroid hormones or inhibiting the absorption of iodine, resulting in hypothyroidism, abnormal thyroid hyperplasia, and goiters [10,11,12,13,14], and fluorine may affect the structure and function of the thyroid as early as childhood [15,16]. In addition, some studies have found that the IQ scores of school-age children living in endemic high-fluoride and high-iodide areas are lower than those of school-age children living in endemic high-fluoride areas or iodine-deficient areas [17]. Our team conducted a preliminary study on the effects of excessive iodine and fluorine on thyroid cells and animal thyroid health [18,19], but at present, there are few studies on human populations, and there are few studies on the combined effects of high iodine and high fluorine intake on children’s intellectual development, and the mechanism is still unclear.

Some studies suggest that fluorine and iodine may have antagonistic effects on the thyroid, and the correlation between TSH and urinary fluoride (UF) is influenced by urinary iodine (UI) [20], that is, fluorine stimulates TH secretion under iodine deficiency and inhibits it under iodine excess [18,21]. At present, it is known that changes in THs are closely related to intelligence [22]. Another hormone that may be related to iodine, fluorine, and intelligence is dopamine (DA) [19]. Excessive iodine can reduce the level of DA in the serum and hippocampus [23], and DA is one of the key substances that is involved in human intelligence and regulates monoamine neurotransmitters [24]. DA can regulate the secretion of TSH and TSH subunits [25]. As an inhibitor of TSH, DA may have an interactive effect on intelligence [19].

Therefore, this study analyzed the effects of different concentrations of iodine and fluoride in drinking water on thyroid health, intelligence, and monoamine neurotransmitters in school-age children, and explored the possible mechanism through which thyroid hormones and monoamine neurotransmitters affect school-age children’s intelligence under the background of high fluorine and iodine intake. Our goal was to provide some epidemiological basis for future research on iodine and fluorine compounds in intelligence and thyroid function.

## 2. Materials and Methods

### 2.1. Study Population and Sampling

In this study, four areas in Tianjin were investigated, and the iodine concentration in water was divided into 300 μg/L [26], where <300 μg/L was a non-iodine endemic area and ≥300 μg/L was an iodine endemic area. According to the Standard for Drinking Water in Rural Areas, the water fluoride concentration is divided into 0.8 mg/L and 1.5 mg/L [27], where <0.8 mg/L is a low fluorine area; 0.8–1.5 mg/L is medium fluorine area; and ≥1.5 mg/L is a high fluorine area. According to the historical data of disease control in Tianjin, Tianjin can be divided into four areas, namely, non-iodine–low-fluorine areas (N-L), non-iodine–medium-fluorine areas (N-M), non-iodine–high-fluorine areas (N-H), and high-iodine–high-fluorine areas (H-H). A total of 399 children aged 7–12 were randomly selected from schools in four regions by cluster sampling. All the respondents signed the informed consent form with their parents, and the ethics committee of Tianjin Center for Disease Control and Prevention passed the ethical argumentation.

The subjects in the four areas were divided into groups according to internal exposure index, urinary iodine, and urinary fluoride concentration. Urinary iodine concentration was divided according to 200 μg/L, with the low urinary iodine group ≤ 200 μg/L and high urinary iodine group > 200 μg/L. Urinary fluoride concentration was divided according to 1.4 mg/L [28], with low urinary fluoride concentration ≤ 1.4 mg/L and high urinary fluoride concentration > 1.4 mg/L. The subjects were divided into four groups: low urinary iodine–low urinary fluoride group (L-L), low urinary iodine–high urinary fluoride group (L-H), high urinary iodine–low urinary fluoride group (H-L) and high urinary iodine–high urinary fluoride group (H-H).

### 2.2. Collection of Drinking Water, Urine and Blood Samples, and Anthropometry

Drinking water samples were collected from villages and schools where the students live. For the centralized water supply, two tap water samples were collected; for the decentralized water supply, a drinking water sample was collected in the east, west, south, north, and middle of the village.

All students collected morning urine for urine iodine and urine fluoride detection. Among them, water iodine and urine iodine were detected by catalytic spectrophotometry of As^3+^-Ce^4+^, which is in line with China’s standard WS/T 107.1-2016 [29]. Fluoride in urine was determined by ion-selective electrode method according to China’s standard WS/T 89-2015 [30].

A total of 5 mL of fasting venous blood was collected, put in an anticoagulant vacuum tube, and stored at −80 °C for later use. The concentrations of free triiodothyronine (FT3), free thyroxine (FT4), and thyroid stimulating hormone (TSH) were determined by the direct chemiluminescence method, which was undertaken by the General Hospital of Tianjin Medical University. The contents of norepinephrine (NE), epinephrine (E), dopamine (DA), and 5-hydroxytryptamine (5-HT) were detected by ELISA kits (Nanjing jiancheng Bioengineering Institute, Nanjing, China, Cat. NO. H096-1-1; H208-1-1; H170-1-1; H104-1-1); microplate reader equipment from Biotek Synergy 2, Winooski, VT, USA.

The height and weight were measured by a standardized method [31], and the height error was controlled within 1 cm and the weight error was controlled within 0.1 kg.

### 2.3. Thyroid Volume Measurement and Thyroid Nodule Examination

A thyroid ultrasound examination was performed by professional operators using an ultrasound machine equipped with a 7.5 MHz transducer. The child was supine and their neck was overstretched. We measured the maximum width (W) on the cross-section and determined the maximum length (L) and depth (D) on the longitudinal section. The formula for calculating thyroid volume is Tvol = 0.479 × (Wleft × Lleft × Dleft + Wright × Lright × Dright) [32]. According to China’s Diagnostic Criteria for Endemic Goiter (WS 276-2007) [33], whether children have goiter or not was judged. The diagnostic criteria are as follows: 7 years old ≤ 4.0 mL, 8 years old ≤ 4.5 mL, 9 years old ≤ 5.0 mL, 10 years old ≤ 6.0 mL, 11 years old ≤ 7.0 mL, and 12 years old ≤ 8.0 mL.

### 2.4. Biochemical Examination of Thyroid Function

According to the description of the direct chemiluminescence method by SIEMENS detection kit, the normal reference ranges of FT3, FT4, and TSH are 3.5–6.5 pmol/L, 11.5–22.7 pmol/L, and 0.64–6.27 mIU/L, respectively. According to previous principles, the following diseases were described [34]: overt hyperthyroidism (TSH < 0.64 mIU/L, FT4 > 22.7 pmol/L, and/or FT3 > 6.5 pmol/L); subclinical hyperthyroidism (TSH < 0.64 mIU/L, FT3 and FT4 normal); overt hypothyroidism (TSH > 6.27 mIU/L, FT4 < 11.5 pmol/L); subclinical hypothyroidism (TSH > 6.27, FT4 normal).

### 2.5. IQ Evaluation

IQ was assessed by the CRT-RC2, which can test IQ [35] and is widely used in intelligence assessment in China, with the advantage of being less influenced by language and ethnic differences [36]. It includes six groups of seventy-two questions: A, AB, B, C, D, and E. All the tests were conducted by trained examiners in the school and completed within 40 min according to the instruction manual. The test results are presented in the form of scores. Among them, a score ≥ 130 is excellent, 120~129 is superior, 110~119 is high normal, 90~109 is normal, 80~89 is dull normal, and 70~79 is marginal.

### 2.6. Investigation of Other Related Factors

We also investigated other possible confounding factors: age, gender, family per capita annual income, parents’ education level, anxiety, tension, irritability, mental trauma, mother’s drinking, smoking and passive smoking during pregnancy, and birth conditions (dystocia, hypoxia, premature delivery, low birth weight, and overdue delivery).

### 2.7. Statistical Analysis

In this study, 399 school-age children were investigated, 395 children received thyroid ultrasound examinations, and 354 blood samples were collected. All the included children lived in this area for at least 3 years, and there was a lack of intelligence test results for one person. P-P test was conducted for the continuous variables surveyed, and descriptive statistics were conducted with the mean standard deviation (SD) for those that conform to the normal distribution, the median, and quartile (P_25_, P_75_) for the skewed variables, and the frequency [proportion (%)] for the classified variables. Variance analysis was used to test the difference between quantitative variables that met the statistical requirements, the nonparametric hypothesis test was used to analyze the difference between skewed quantitative variables, and the chi-square test was used to analyze the difference between classified variables. Univariate and multivariate piecewise linear regression and multivariate logistic regression were used to analyze the relationship between thyroid hormones, intelligence, thyroid health status, and monoamine neurotransmitters in different areas of water fluoride–water iodide concentrations and urine fluoride–urine iodide grouping.

The restricted cubic spline (RCS) was used to analyze the linear relationship between urine fluoride–urine iodine and water fluoride–water iodine and thyroid hormones, intelligence, thyroid health status, and monoamine neurotransmitters, which were divided into two models: (1) a rough model without considering other factors; (2) a model adjusted according to the confounding factors of multiple linear regression and multiple logistic regression. Path analysis was used to explain the mediating effect of thyroid hormone and monoamine neurotransmitters in the relationship between urinary fluoride–urinary iodine concentration and IQ. RCS was used to analyze the relationship between urinary iodine and urinary fluoride, thyroid nodules, and thyroid hormones. Because the number of dependent variable samples was more than 100, the number of spline function nodes was selected as 5, the tangent point of normal variables was selected as the average, and the tangent point of skewness and classification variables were selected as the median [37]. In all analyses, the statistical significance level *α* of the two-sided test was set to 0.05. The potential interaction between urinary iodine and urinary fluoride was analyzed by adding and multiplying the interaction model. We calculated the relative excess risk of interaction (RERI) and attributable proportion (AP); if the 95% CI of RERI and AP contained 0, there was no interaction [38,39].

The data analysis used R version 4.3.0 (http://www.r-project.org, accessed on 27 December 2023) and SPSS 24.0 (IBM, Chicago, IL, USA) software.

## 3. Results

### 3.1. Characteristics of the Participants

A total of 399 children were investigated in this study, including 201 boys (50.38%) and 198 girls (49.62%). As shown in Table 1, the age range of children is 9.47 ± 1.15 years; the BMI range is 16.74 (15.23, 18.90) kg/m^2^; the urine iodine concentration is 229.30 (160.65, 337.20) μg/L; the urine fluoride concentration is 1.29 (0.89, 1.84) mg/L; and the volume of thyroid gland is 1.26 (1.02, 2.24) mL. THs concentrations were as follows: TSH: 2.75 (2.06, 3.77) μIU/mL; FT3: 6.72 ± 0.76 pmol/L; FT4: 16.88 ± 1.88 pmol/L. IQ was 112.16 ± 13.55. The concentrations of monoamine neurotransmitters, shown in Table 1 and Appendix A are as follows: 5-hydroxytryptamine (5-HT): 127.44 (79.08, 245.91) ng/mL; norepinephrine (NE): 1094.07 (853.77, 1366.29) ng/L; dopamine (DA): 5.77 (2.86, 11.54) mg/mL; and epinephrine (E): 350.30 (268.68, 517.98) ng/L. School-age children in different iodine–fluorine areas suffer from thyroid nodules, goiter, intelligence level, thyroid health, drinking water types, seafood, weekly passive smoking frequency, weekly exercise frequency, stress, passive smoking frequency of mothers during pregnancy, family income, and father’s education level. There were significant statistical differences in age, FT3, FT4, Tvol, IQ, UI, UF, thyroid nodules, goiter, thyroid health, drinking water types, and head and neck treatment history of school-age children in different urinary iodine–urinary fluoride groups (*p* < 0.05). See Table 1 and Appendix A.

### 3.2. Effect of Different Water Iodine and Fluorine Areas on Thyroid Health of School-Age Children

Compared with the N-L control group, there was no statistical difference in thyroid health indices in the N-M group. The N-H group showed significant protective effects on thyroid nodules (crude OR: 0.12 (0.01~0.94)) and thyroid health abnormalities (adjusted OR: 0.33 (0.12~0.93)), especially on Tvol, FT4, and TSH (adjusted *β*: −0.31 (−0.50~−0.12), −1.08 (−1.64~−0.53), and −0.66 (−1.15~−0.17)). Thyroid nodules and abnormal thyroid health (adjusted OR: 5.27 (1.84~15.12); 5.68 (2.63~12.25)) showed a significant high risk in H-H group, Tvol (adjusted *β*: 2.96 (2.75~3.16)) showed a significant positive correlation, and FT3 (adjusted *β*: −0.38 (−0.62~−0.14)) level showed a significant positive correlation, see Table 2.

### 3.3. Influence of Different Water Iodine and Water Fluoride Areas on the Intelligence of School-Age Children

There were significant differences in serum 5-HT and DA among different groups (*p* < 0.01, see Table 1). Compared with the N-L group, the 5-HT in the H-H group showed a significant negative correlation, and the adjusted β was −55.61 (95% CI: −104.67, −6.54). In the N-M and N-H groups, the concentration of DA showed a significant positive correlation, and the adjusted *β* was 3.34 (0.78~5.89) and 3.08 (0.42~5.75), respectively (see Table 3).

The intelligence grades of school-age children are different in each group, which is mainly manifested in that the ratio of outstanding intelligence respondents with 110 points or more in N-H group (13.78%) and H-H group (10.03%) is lower than that in N-L group (16.79%) and N-M group (16.54%). Compared with N-L group, the adjusted *β* of IQ in the H-H group is −5.85 (95% CI: −10.13, −1.57) (see Table 3), and the IQ score in the H-H group is significantly different from other groups (adjusted *p* < 0.05) (see Appendix A).

### 3.4. Relationship between Urinary Iodine, Urinary Fluorine, and Thyroid Health of School-Age Children

The Tvol of the H-H group was significantly different from other groups (adjusted *p* < 0.01), which was 2.67 (P_25_, P_75_: 1.22,4.26)mL, and compared with the L-L group, the adjusted *β* was 1.55 (95% CI: 1.21, 1.89). The proportion of patients with thyroid nodules (6.52%), goiter (3.26%), and thyroid abnormality (11.78%) in the H-H group was also significantly different (adjusted *p* < 0.05). Compared with the L-L group, the adjusted OR of thyroid nodule and thyroid abnormality was 2.85 (95% CI: 1.22, 6.69) and 2.99 (95% CI: 1.56, 5.71). The concentration of FT3 in the H-H group was 6.47 ± 0.80 pmol/L, the lowest in all groups, and the difference was significant (adjusted *p* < 0.01). Compared with the L-L group, the adjusted *β* is −0.35 (95% CI: −0.53, −0.16) pmol/L (see Table 4 and Appendix A).

In the RCS rough model, Tvol increased with the increase in urinary iodine concentration when urinary iodine concentration was more than 227.59 μg/L, and reached a plateau after urinary iodine concentration was about more than 500 μg/L, and this trend was statistically significant (overall *p* < 0.001, see Figure 1A). When UI < 234.01 μg/L, the OR value of the thyroid nodules first increased and then decreased. When UI > 234.01 μg/L, there was a significant positive correlation between urinary iodine concentration and the prevalence of thyroid nodules (overall *p* < 0.001, see Figure 1B). At 225.12 μg/L < UI < 480.37 μg/L, TSH concentration was positively correlated with urinary iodine, while at UI > 480.37 μg/L, TSH concentration was negatively correlated with urinary iodine, and the nonlinear relationship was statistically significant (nonlinear *p*: 0.045, Figure 1C). When UI > 158.97 μg/L, FT3 concentration was negatively correlated with UI concentration (overall *p* < 0.001, Figure 1D). In the rough model, when the urine fluoride is more than 1.29 mg/L, Tvol is positively correlated with the urine fluoride concentration, and the trend is similar to that of urine iodine (overall *p* < 0.001, Figure 1E). When urinary fluoride is greater than 1.30 mg/L, the OR value of the thyroid nodules is positively correlated with urinary fluoride concentration, and its influence trend is similar to that of urinary iodine (overall *p*: 0.003, Figure 1F). When UF > 1.49 mg/L, FT3 concentration was negatively correlated with urine fluoride concentration (overall *p*: 0.008, Figure 1G). See Appendix A for more data.

After adjusting for confounding factors, Tvol still showed a significant positive correlation when UI was higher than 228.07 μg/L, and Tvol had a strong correlation with iodine (overall *p* < 0.001, Figure 2A). When UI > 154.20 μg/L, FT3 concentration is still negatively correlated with UI concentration and FT3 concentration is also strongly correlated with iodine (overall *p*: 0.038, Figure 2B). After adjusting for confounding factors, when UF > 1.49 mg/L, Tvol was still positively correlated with urine fluoride concentration (overall *p* < 0.001, see Figure 2C). After adjusting for confounding factors, the relationship between the thyroid nodules and FT3 and UF lost statistical significance but FT4 showed a significant negative correlation when UF > 1.49 mg/L (overall *p*: 0.017, Figure 2D). See Appendix A for more data.

### 3.5. Relationship between Urinary Iodine, Urinary Fluoride, and Intelligence of School-Age Children

Compared with the L-L group, the IQ of school-age children in the L-H group and the H-H group showed significant differences. The intelligence scores of the latter two groups were 107.86 ± 13.78 and 110.90 ± 13.65, and the *β* was −6.72 (95% CI: −12.87, −0.58) and−3.68 (95% CI: −7.04, −0.33) (see Table 5).

In the RCS rough model, when UI > 291.33 μg/L, iodine has a significant negative effect on intelligence (overall *p*: 0.047, see Figure 3A); when UF > 1.47 mg/L, fluorine has some negative effects on intelligence (overall *p*: 0.075, see Figure 3B). After adjusting for confounding factors, the influence of iodine on intelligence is significantly reduced (overall *p*: 0.077, see Figure 3C), and the influence of fluorine on intelligence is significantly enhanced. When UF > 1.28 mg/L, fluorine has a significant negative impact on school-age children’s intelligence (overall *p*: 0.064, see Figure 3D). See Appendix A for more data.

In the RCS rough model, the relationship between both urinary iodine and urinary fluoride and four monoamine neurotransmitters was not statistically significant. After adjusting for confounding factors, when UI > 228.50 μg/L, the dopamine concentration decreased with the increase in iodine concentration in urine (overall *p*: 0.045, see Figure 4A); when UF > 1.27 mg/L, the dopamine concentration and fluoride concentration in urine showed a “U”-shaped relationship (overall *p*: 0.030, see Figure 4C). There is an inverted “U” relationship between 5-HT and urinary iodine concentration, which reaches the highest concentration at 291.33 μg/L, and high iodine affects the decrease in 5-HT secretion (overall *p*: 0.087, see Figure 4B). See Appendix A for more data.

### 3.6. The Role of Thyroid Hormones and/or Monoamine Neurotransmitters with the Influence of Iodine and Fluorine on the Intelligence and Thyroid Health of School-Age Children

In this investigation, the combined action of iodine and fluorine has a major health impact on school-age children, including intellectual impairment and increased risk of thyroid diseases, especially goiter and thyroid nodules. The biochemical indices investigated in this study mainly include the secretion level of thyroid hormone and the serum content of monoamine neurotransmitters in school-age children. To further analyze the role of thyroid hormone and monoamine neurotransmitters in the above health damage that may be caused by high iodine and high fluorine, path analysis is used to model, but the mixed factors in the questionnaire are not considered in the model, and only the relationship between the biochemical indices and the outcome of this survey is observed.

The results of path analysis show that iodine and fluorine affect thyroid hormones, mainly, iodine affects FT3 and negatively regulates FT3 secretion, which is consistent with the results of the previous RCS analysis. There is a positive correlation between FT3 and FT4, and there is a negative feedback regulation correlation between FT4 and TSH. Iodine causes the above three thyroid hormones to change, and all three thyroid hormones affect the evaluation results of children’s intelligence. Among them, FT3 and TSH negatively affect IQ, while FT4 positively affects IQ. When the concentration of thyroid hormone in the blood changes, it may affect the secretion of neurotransmitters through the hypothalamus–pituitary–thyroid axis. The results of path analysis show that FT4 can positively affect the secretion of DA, NE, and 5-HT, while FT3 mainly positively affects the secretion of DA. Adrenaline is not included in model optimization but the secretion of adrenaline is closely related to norepinephrine, and the correlation between them is included in the model. However, the results of path analysis show that the four monoamine neurotransmitters in this investigation have no significant influence on IQ, but this cannot rule out the intermediary role of monoamine neurotransmitters in the intellectual damage of school-age children caused by the combined action of iodine and fluorine (see Figure 5A and Appendix A). To sum up, the influence of iodine and fluorine on the thyroid intelligence of school-age children may be shown in Figure 5B.

The results of the path analysis showed that urinary iodine concentration positively affected thyroid volume, negatively affected FT3 hormone secretion, and FT4 hormone positively affected Tvol. In this model, there was a significant positive correlation between Tvol and thyroid nodules, and also a significant positive correlation between urinary iodine and urinary fluoride, FT3 and FT4, and a significant negative correlation between FT4 and TSH. In addition, the fitted model showed the iodine and fluorine pathways to thyroid health in school-age children under the combined influence of iodine and fluorine. See Figure 5C,D, and Appendix A.

### 3.7. Interaction between Iodine and Fluorine on Thyroid and Intelligence of School-Age Children

After analyzing the interaction between urinary fluoride and urinary iodine on thyroid nodules, goiter, intelligence level, IQ, Tvol, TSH, FT3, FT4, NE, 5-HT, DA, and E of school-age children by the addition and/or multiplication interaction model, the results show that urinary iodine and urinary fluoride have interactive effects on the thyroid nodules and IQ of school-age children. In the additive model, the OR of UI and UF interaction is 3.19 (95% CI: 1.38–7.36) and 1.01 (1.00–1.01), and the RERI is 1.37 (0.74–2.01) and 0.97 (0.82~1.13). The AP values were 3.08 (0.32~5.83) and 0.97 (0.82~1.11), indicating that UI and UF may have additive synergistic effects on the thyroid nodules and IQ of children with academic qualifications, as shown in Table 6.

## 4. Discussion

Both iodine and fluorine have certain effects on the human thyroid, and there is an interaction between them [9,40]. In this study, it was also found that there was a significant correlation between urinary iodine and urinary fluoride, and the area with high fluorine and high iodine had a significant impact on THs secretion and thyroid health. Many previous studies have shown that iodine is very important to thyroid volume [32,41,42], there is a “U”-shaped nonlinear relationship between iodine and Tvol. When the iodine concentration exceeds a certain threshold, it may cause goiter in children [32]. At present, goiter caused by iodine excess may be caused by negative feedback caused by iodine regulation of THs [43]. The increase in iodine load induces the acute Wolff–Chaikoff effect, which leads to a decrease in THs’ synthesis. When THs decrease, thyrotropin increases, which stimulates the diffuse proliferation of thyroid follicles and eventually leads to goiter [43,44]. In this study, it was found that high-water iodine had a greater impact on the thyroid gland. In the environment of excessive fluorine in water, the risk of thyroid nodules and thyroid abnormalities in school-age children increased by about 43 and 17 times, respectively. For every unit of iodine in water, Tvol increased by 2.96 mL on average, while FT4 and TSH secretion did not change significantly, only FT3 secretion decreased by 0.38 pmol/L on average. There may be some antagonism between water iodine and fluorine in the thyroid, and the influence of fluorine on the thyroid may reduce the influence of iodine. In the case of excessive iodine in water, the risk of thyroid nodules and thyroid abnormalities decreased by 0.12 and 0.33 times for every unit of fluorine in water, while Tvol, FT4 and TSH decreased by 0.31 mL, 1.08 pmol/L, and 0.66μIU/mL on average but the effect was weak, which may affect the activity of TSH by inhibiting the activity of thyroidic acid cyclase [45,46]. According to a clinical study, TSH > 2.5 μIU/mL is an independent risk factor for thyroid nodules in children [47]. Generally speaking, most thyroid nodules are benign; however, when children or adolescents are found to have thyroid nodules, compared with adults, the risk of malignant tumors increases by 2–3 times [48,49]. In addition, iodine and fluorine may also affect the hypothalamus–pituitary–thyroid hormone axis [50,51] and the development of the pituitary–TH axis in children and adolescents, which also participates in the occurrence and development of thyroid nodules [52,53]. There is an interesting phenomenon in drinking water types. In areas with high iodine and high fluorine, the main drinking way is tap water, while in non-high iodine or non-high fluorine areas, the main drinking way is bottled drinking water, while in areas suitable for water iodine and low fluorine, the respondents eat more seafood. After adjusting the above confounding factors in the N-H group, the influence of high fluorine on thyroid nodules has lost its significance, which shows that such a lifestyle may aggravate the thyroid health of the respondents. Because the concentration of iodine and fluorine in urine is positively correlated with the intake of iodine and fluorine in water [28,54], in this study, the concentration of iodine and fluorine in urine was used as the internal exposure index. It was found that the Tvol volume increased by about 1.53 mL for every unit of urine iodine and urine fluoride, and the possibility of thyroid nodules or abnormalities in school-age children with high urine iodine and high urine fluoride was 1.78 and 1.98 times higher than those with low urine iodine and low urine fluoride. When iodine and urine fluoride increase by one unit on average, FT3 concentration decreases by about 0.35 pmol/L, and the thyroid volume in the group with high urine iodine and urine fluoride is about twice as high as that in other groups, indicating that iodine and fluorine contribute to the increase in thyroid volume, which is consistent with the result of water iodine and water fluoride. The concentration of FT3 hormone in the high urinary iodine–high urinary fluoride group was also lower than that in other groups, which was consistent with the result of water iodine–water fluoride. The patients with thyroid nodules and abnormal thyroid were mainly concentrated in the high urinary iodine–high urinary fluoride group. This study provides a certain epidemiological basis for the combined effects of iodine and fluorine on THs and thyroid health in children and adolescents, and the effects of iodine and fluorine on thyroid diseases in children need further research on their mechanisms.

However, what deserves more attention in this study is the combined effect of iodine and fluorine on the intelligence of school-age children. By fitting the nonlinear relationship between urine iodine, urine fluoride, and intelligence, it is found that iodine, fluorine, and intelligence evaluation scores of school-age children are indeed negatively correlated as previously assumed. Compared with other regions, the intelligence scores of respondents in areas with high water iodine and high water fluoride are lower, and more respondents are concentrated in the normal file, while the number of respondents with higher IQ is reduced. As every unit of water fluoride and water iodine increases, IQ decreases by 5.85 points, which is consistent with many previous related research conclusions [19,26,40,55,56]. Among them, the influence of fluorine exposure on school-age intelligence is mainly manifested in its threshold and saturation effect, and exposure to moderate excessive fluorine is mainly related to the loss of excellent intelligence [26]. Fluorine can penetrate the brain through the blood–brain barrier [57], and the accumulated fluorine will cause learning function decline and memory defects [58], which will lead to nerve damage in the central system and change cognition, behavior, and neuropsychiatry [26,59]. In this study, the role of monoamine neurotransmitters in the above-mentioned influencing pathways was discussed. It has been found that DA-related genes may modify the association between urinary fluoride and intelligence assessment, mainly ANKK1 Taq1A, COMT Val 158 Met, and MAOA uVNTR show high-dimensional interaction with IQ [56]. DA is widely considered to be involved in the learning, memory, fluid intelligence development, and cognitive plasticity of school-age children [60,61]. Fluorine can affect the concentration of DA in the striatum and cerebellum of rats, and lead to a change in dopamine receptor expression [62]. In the model in this study, the concentration of DA in the serum of school-age children is positively correlated with the concentration of fluorine in water in non-high iodine areas. The relationship between urinary fluoride and DA is that THs play an intermediary role in it, and it is speculated that changing the path may be a way to influence school-age children. However, unlike fluorine, the influence of iodine on the intelligence of school-age children is more complicated, and both iodine deficiency and excess may aggravate the negative effect of fluorine on intelligence [17,21]. It is reported that iodine deficiency may aggravate thyroid diseases caused by fluorine [16], and some studies suggest that high levels of iodine may reduce the effect of fluorine on the thyroid [9,18].

In addition, this study found that THs have a direct impact on the intelligence of school-age children, among which TSH and FT3 are negatively correlated with IQ, while FT4 is positively correlated with IQ. Elevated serum TSH levels can diagnose primary hypothyroidism, and a low FT4 concentration with a normal or low serum TSH level indicates hypothalamic–pituitary hypothyroidism, and hypothalamic–pituitary–thyroid function will affect children’s intellectual development [63]. Some studies have also found that the direct correlation between TSH and IQ is not significant [40,64] but low FT4 is indeed related to neurodysplasia in children [65]. T3 is a TH with higher activity. In normal thyroid, 90% of thyroid hormone receptors are occupied by T3 [66], while the level of FT3 in a group with adolescent bipolar disorder is even lower. The influence of thyroid hormone on IQ is probably through the hypothalamic–pituitary–gonadal (HPG) axis [67,68]. In addition, THs can affect the functions of serotonin and catecholamine in the brain, and the plasma serotonin level is positively correlated with FT3 concentration [69]. Animal studies have shown that THs can increase serotonin in the cerebral cortex through 5-HT [70], and other neurotransmitters are also affected by thyroid hormones, including dopamine receptors and signal transduction processes, as well as gene regulation mechanisms [71]. The noradrenergic neurotransmission activity of some receptors decreases in the state of hypothyroidism, and the increase in THs can also increase their ability to accept stimuli [72,73]. The conclusion of the above study is basically consistent with that of this study, that is, the changes in thyroid hormone caused by iodine and fluorine affect the level of monoamine neurotransmitters in the serum of school-age children; however, the influence of monoamine neurotransmitters on school-age children’s emotion, intelligence, and cognition may be mainly realized through the hypothalamus–pituitary–gonad (HPG) and hypothalamus–pituitary–adrenal (HPA) axes [68]. Although the path analysis of monoamine neurotransmitters in this study does not show a direct impact on IQ, it will indirectly affect the evaluation results of intelligence by affecting the cognition and emotion of school-age children [70,71].

However, there are some limitations of this study. Firstly, as this study only explored the combined effects of iodine and fluorine on thyroid, monoamine neurotransmitters, and IQ in schoolchildren using some biochemical endo-exposure indices and questionnaires, the possible pathways of action and nonlinear relationships were obtained after statistical analysis and modeling but it was not possible to explore the underlying mechanisms of these relationships. Further studies in the future could explore the possible mechanisms of action of iodine and fluorine through in vitro and in vivo experiments. Secondly, the data in this study were mainly derived from a cross-sectional study in Tianjin, China, and it is possible that the interactions of the effects of iodine and fluorine on thyroid health and intelligence in school-aged children may be different in other regions due to different intakes of iodine and fluorine. In the future, it is hoped that more survey data from different regions will enrich this science field.

## 5. Conclusions

The main effects of high iodine and high fluorine in water on the health of school-age children are reflected in intelligence and thyroid health. The influence on thyroid health may mainly lead to a change in thyroid capacity through the storage and release of iodine and fluorine by the thyroid, which in turn leads to a change in thyroid hormones and an increased risk of thyroid nodules and goiter. The influence on intelligence may be mainly related to the above-mentioned changes in the thyroid gland. Iodine and fluorine affect the intelligence evaluation of school-age children directly or indirectly through monoamine neurotransmitters by affecting the changes in the thyroid gland and thyroid hormones, and affect the results of the intelligence evaluation of school-age children directly or indirectly through regulating thyroid hormones and monoamine neurotransmitters through HPG and HPA axes through the blood–brain barrier.

## Figures and Tables

**Figure 1 nutrients-16-02913-f001:**
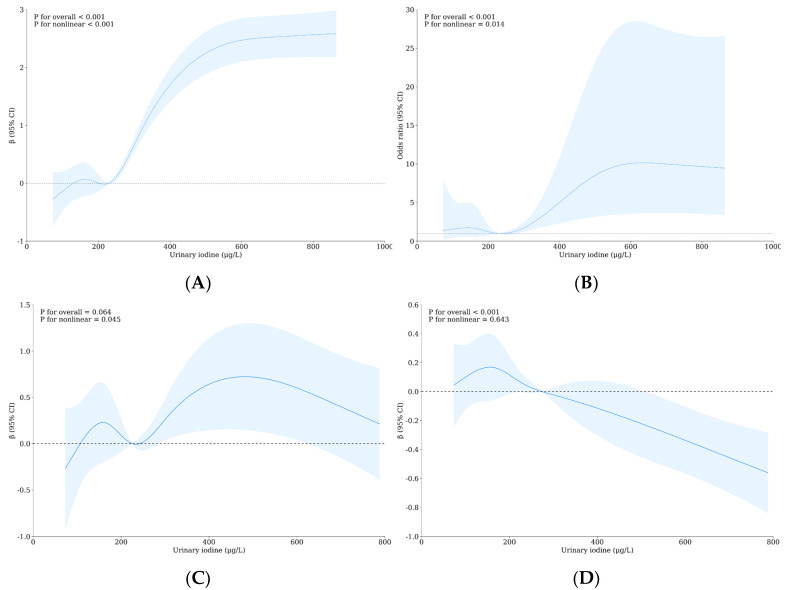
Coarse-model nonlinear relationships between urinary iodine with urinary fluoride concentrations and thyroid health. (**A**) The relationship between Tvol and UI; (**B**) the relationship between thyroid nodules and UI; (**C**) the relationship between TSH and UI; (**D**) the relationship between FT3 and UI; (**E**) the relationship between Tvol and UF; (**F**) the relationship between thyroid nodules and UF; (**G**) the relationship between FT3 and UF.

**Figure 2 nutrients-16-02913-f002:**
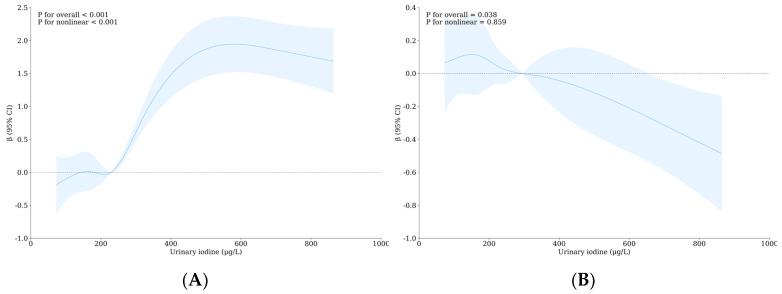
Nonlinear relationship between urinary iodine with urinary fluoride concentration and thyroid health after adjusting for confounders. (**A**) The relationship between Tvol and UI after adjusting confounding factors; (**B**) The relationship between FT3 and UI after adjusting confounding factors; (**C**) the relationship between Tvol and UF after adjusting confounding factors; (**D**) the relationship between FT4 and UF after adjusting confounding factors.

**Figure 3 nutrients-16-02913-f003:**
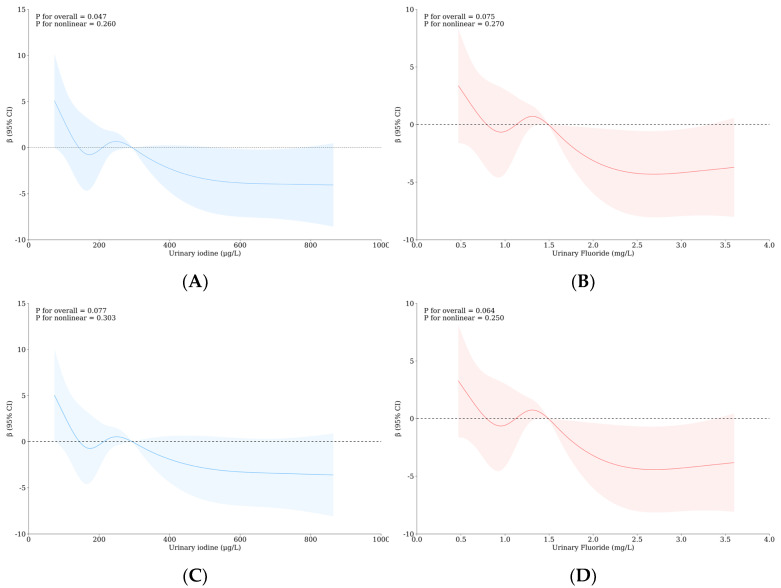
Nonlinear relationship between urinary iodine as well as urinary fluoride concentrations and intelligence. (**A**) The relationship between IQ and UI after adjusting confounding factors; (**B**) the relationship between IQ and UF after adjusting confounding factors; (**C**) the relationship between IQ and UI after adjusting confounding factors; (**D**) the relationship between IQ and UF after adjusting confounding factors.

**Figure 4 nutrients-16-02913-f004:**
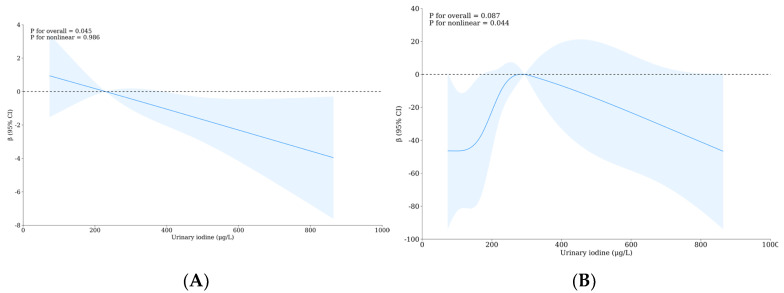
Nonlinear relationship between both urinary iodine and urinary fluoride concentrations and monoamine neurotransmitters. (**A**) The relationship between DA and UI after adjusting confounding factors; (**B**) the relationship between 5-HT and UI after adjusting confounding factors; (**C**) the relationship between DA and UF after adjusting confounding factors.

**Figure 5 nutrients-16-02913-f005:**
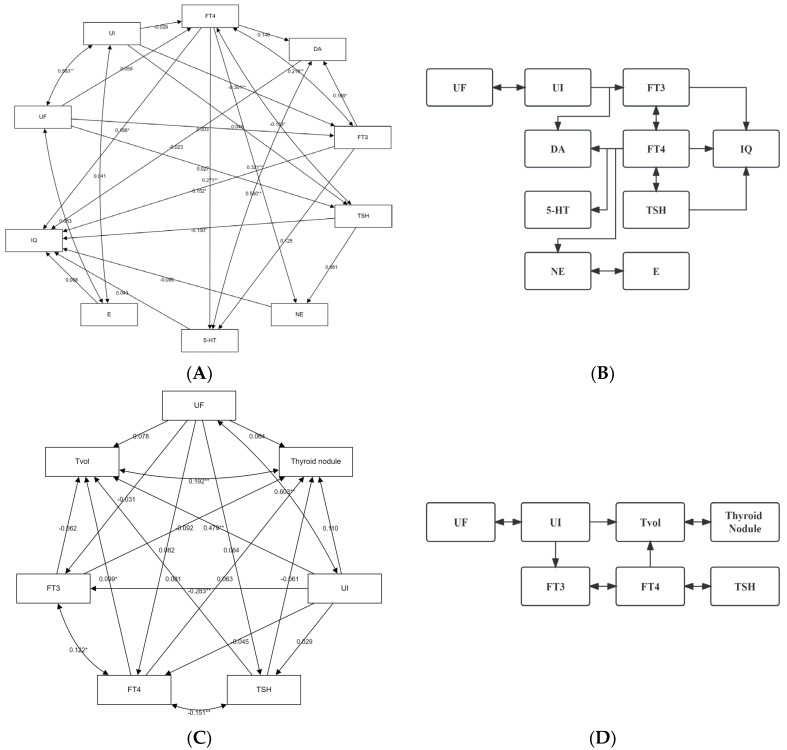
Pathway analysis of thyroid hormones and/or monoamine neurotransmitters combined with iodine- and fluorine-induced mental impairment and thyroid disease. (**A**) Path analysis model diagram of urinary fluoride–urinary iodine, thyroid hormone, monoamine neurotransmitter, and intelligence; (**B**) the possible ways iodine and fluorine jointly influence the intelligence of school-age children through thyroid hormones and/or monoamine neurotransmitters; (**C**) path analysis model diagram of the influence of urinary fluoride–urinary iodine and thyroid hormone on thyroid health; (**D**) possible paths of iodine and fluorine through thyroid hormones or directly affecting the thyroid health of school-age children. * *p* < 0.05; ** *p* < 0.01.

**Table 1 nutrients-16-02913-t001:** Basic characteristics of the study population.

Variable	Level	WI-WF Group ^a, b^				*p*	Overall	UI-UF Group ^a, b^				*p*	Overall
		N-H	N-L	N-M	H-H			L-L	L-H	H-L	H-H		
Sample size		100	99	100	100		399	115	22	93	135		365
Age (mean ± SD)	years old	9.35 ± 1.21	9.24 ± 1.26	9.58 ± 1.22	9.72 ± 0.79	0.013 *	9.47 ± 1.15	9.40 ± 1.31	9.82 ± 1.15	9.26 ± 1.17	9.64 ± 0.97	0.036 *	9.48 ± 1.16
FT3 (mean ± SD)	pmol/L	6.86 ± 0.62	6.70 ± 0.85	6.87 ± 0.80	6.34 ± 0.65	<0.001 **	6.72 ± 0.76	6.83 ± 0.65	6.95 ± 0.42	6.82 ± 0.63	6.47 ± 0.80	<0.001 **	6.71 ± 0.71
FT4 (mean ± SD)	pmol/L	16.28 ± 1.53	17.36 ± 2.00	16.86 ± 1.92	17.07 ± 1.92	<0.001 **	16.88 ± 1.88	16.79 ± 1.78	17.41 ± 1.71	16.48 ± 1.96	17.16 ± 2.00	0.047 *	16.88 ± 1.91
IQ (mean ± SD)	scores	112.07 ± 15.50	114.04 ± 11.05	114.22 ± 11.95	108.33 ± 14.51	0.006 **	112.16 ± 13.55	114.59 ± 13.73	107.86 ± 13.78	112.18 ± 12.76	110.90 ± 13.65	0.070	112.20 ± 13.54
BMI (median [IQR])	kg/m^2^	17.12 (15.75, 20.5)	16.89 (15.44, 18.31)	16.22 (14.54, 18.90)	16.35 (14.83, 18.56)	0.020 *	16.74 (15.23, 18.90)	16.96 (15.46, 19.14)	15.76 (14.37, 19.38)	16.73 (15.33, 18.65)	16.35 (15.02, 19.08)	0.515	16.74 (15.22, 18.90)
Tvol (median [IQR])	mL	0.99 (0.88, 1.13)	1.29 (1.20, 1.46)	1.14 (0.93, 1.30)	4.08 (3.38, 4.95)	<0.001 **	1.26 (1.02, 2.24)	1.16 (1.02, 1.36)	1.01 (0.94, 1.47)	1.23 (1.08, 1.69)	2.67 (1.22, 4.26)	<0.001 **	1.27 (1.02, 2.40)
TSH (median [IQR])	μIU/mL	2.73 (1.98, 3.28)	2.97 (2.20, 4.16)	2.64 (2.19, 3.69)	2.71 (1.75, 4.32)	0.167	2.75 (2.06, 3.77)	2.64 (2.16, 3.64)	2.91 (2.26, 3.53)	2.84 (2.18, 3.96)	2.80 (2.05, 4.08)	0.740	2.77 (2.09, 3.75)
UI (median [IQR])	μg/L	187.65 (133.40, 234.15)	217.20 (158.38, 289.73)	209.15 (137.68, 255.12)	476.30 (333.45, 629.45)	<0.001 **	229.30 (160.65, 337.20)	139.50 (110.10, 162.65)	167.15 (135.90, 183.43)	253.50 (222.50, 308.90)	376.00 (270.80, 525.45)	<0.001 **	227.50 (159.70, 336.30)
UF (median [IQR])	mg/L	1.01 (0.75, 1.40)	1.05 (0.76, 1.52)	1.24 (0.89, 1.47)	2.18 (1.52, 2.79)	<0.001 **	1.29 (0.89, 1.84)	0.89 (0.71, 1.07)	1.62 (1.46, 1.89)	1.01 (0.83, 1.24)	2.10 (1.64, 2.70)	<0.001 **	1.29 (0.89, 1.86)
5-HT (median [IQR])	ng/mL	125.43 (78.84, 253.54)	139.39 (97.00, 334.83)	134.10 (83.37, 232.35)	104.45 (70.79, 202.08)	0.036 *	127.44 (79.08, 245.91)	114.81 (74.45, 241.98)	123.17 (70.33, 218.88)	158.76 (87.49, 300.60)	124.04 (79.93, 230.67)	0.200	128.16 (79.77, 245.91)
NE (median [IQR])	ng/L	1169.83 (854.96, 1369.12)	1184.24 (885.91, 1588.78)	1008.21 (808.65, 1281.21)	1078.17 (924.48, 1311.28)	0.124	1094.07 (853.77, 1366.29)	1107.81 (842.60, 1305.38)	1249.23 (943.59, 1483.40)	1006.82 (7714.1, 1329.36)	1119.66 (923.62, 1413.85)	0.244	1094.19 (854.78, 1367.70)
DA (median [IQR])	mg/mL	7.34 (4.80, 13.28)	4.21 (2.54, 8.82)	6.06 (2.88, 12.64)	4.57 (1.95, 9.48)	0.004 **	5.77 (2.86, 11.54)	5.95 (3.74, 11.68)	6.99 (3.99, 12.26)	6.84 (2.93, 14.37)	5.32 (2.65, 9.79)	0.299	5.98 (3.04, 11.44)
E (median [IQR])	ng/L	334.15 (286.60, 488.81)	347.47 (259.41, 487.11)	356.00 (255.79, 496.39)	386.13 (284.82, 624.17)	0.240	350.30 (268.68, 517.98)	318.08 (258.29, 502.60)	402.09 (245.86, 585.47)	334.98 (284.33, 467.08)	391.74 (265.15, 561.89)	0.475	345.55 (266.33, 522.50)
Gender (%)	Boys	50 (50.00)	48 (48.48)	50 (50.00)	53 (53.00)	0.934	201 (50.38)	64 (55.65)	14 (63.64)	45 (48.39)	70 (51.85)	0.535	193 (52.88)
	Girls	50 (50.00)	51 (51.52)	50 (50.00)	47 (47.00)		198 (49.62)	51 (44.35)	8 (36.36)	48 (51.61)	65 (48.15)		172 (47.12)
Thyroid nodule (%)	No	99 (99.00)	91 (91.92)	96 (96.00)	73 (73.00)	<0.001 **	359 (89.97)	107 (93.04)	22 (100.00)	90 (96.77)	109 (80.74)	<0.010 **	328 (89.86)
	Yes	1 (1.00)	8 (8.08)	4 (4.00)	27 (27.00)		40 (10.03)	8 (6.96)	0 (0.00)	3 (3.23)	26 (19.26)		37 (10.14)
Goiter (%)	Yes	0 (0.00)	0 (0.00)	0 (0.00)	15 (15.00)	<0.001 **	15 (3.76)	0 (0.00)	0 (0.00)	1 (1.08)	13 (9.63)	<0.010 **	14 (3.84)
	No	100 (100.00)	99 (100.00)	100 (100.00)	85 (85.00)		384 (96.24)	115 (100.00)	22 (100.00)	92 (98.92)	122 (90.37)		351 (96.16)

** p* < 0.05, ** *p* < 0.01; ^a^ Statistical value: ANOVA: *F*-value—age, FT3, FT4, IQ; Kruskal–Wallis test: *H*-value—BMI, Tvol, TSH, UI, UF, 5-HT, NE, DA, E; other factors: *χ*^2^ value; ^b^ WI-WF group: the subjects were grouped according to different concentrations of iodine and fluoride in water; UI-UF group: the subjects were divided into groups according to different urine iodine and urine fluoride concentrations; see Section 2.1 for details.

**Table 2 nutrients-16-02913-t002:** Effects of water iodine and water fluoride on thyroid health in schoolchildren.

Factor	WI-WF Group ^d^					
	N-M ^b^		N-H ^b^		H-H ^b^	
	Crude OR/ *β* (95% CI) ^a^	Adjusted OR/ *β* (95% CI) ^a,c^	Crude OR/ *β* (95% CI) ^a^	Adjusted OR/ *β* (95% CI) ^a,c^	Crude OR/ *β* (95% CI) ^a^	Adjusted OR/ *β* (95% CI) ^a,c^
Thyroid nodule	0.47 (0.14~1.63)	0.50 (0.13~1.92)	0.12 (0.01~0.94) *	0.12 (0.01~1.15)	4.21 (1.80~9.81) *	5.27 (1.84~15.12) *
Thyroid abnormal	0.56 (0.25~1.25)	0.55 (0.23~1.33)	0.29 (0.11~0.76) *	0.33 (0.12~0.93) *	4.32 (2.27~8.23) *	5.68 (2.63~12.25) *
Thyroid disease	0.98 (0.65~1.47)	1.07 (0.57–2.03)	1.28 (0.79~2.08)	1.08 (0.58–2.00)	0.84 (0.57~1.23)	1.06 (0.56–2.02)
Tvol	−0.16 (−0.35~0.03)	−0.15 (−0.35~0.04)	−0.27 (−0.46~−0.08) *	−0.31 (−0.50~−0.12) *	2.91 (2.72~3.10) *	2.96 (2.75~3.16) *
FT3	0.17 (−0.04~0.39)	0.18 (−0.04~0.40)	0.16 (−0.05~0.38)	0.15 (−0.07~0.37)	−0.36 (−0.58~−0.13) *	−0.38 (−0.62~−0.14) *
FT4	−0.50 (−1.03~0.02)	−0.39 (−0.94~0.17)	−1.09 (−1.62~−0.56) *	−1.08 (−1.64~−0.53) *	−0.30 (−0.86~0.27)	−0.23 (−0.84~0.37)
TSH	−0.35 (−0.82~0.12)	−0.32 (−0.81~0.17)	−0.61 (−1.08~−0.14) *	−0.66 (−1.15~−0.17) *	−0.10 (−0.60~0.40)	−0.06 (−0.59~0.47)

* *p* < 0.05; ^a^. OR: thyroid nodule, goiter, thyroid abnormal, thyroid disease; *β:* Tvol, FT3, FT4, TSH; ^b^. N-L group served as the control group; ^c^. Mixed factors of adjustment: the confounding factors of thyroid nodule, goiter, thyroid normal, thyroid disease, Tvol, FT3, FT4, and TSH adjusted Drinking water type, Seafood, Age, and BMI; ^d^. WI-WF group: the subjects were grouped according to different concentrations of iodine and fluoride in water; see Section 2.1 for details.

**Table 3 nutrients-16-02913-t003:** Effects of water iodine and water fluoride on the intelligence of schoolchildren.

Factor	WI-WF Group ^d^					
	N-M ^b^		N-H ^b^		H-H ^b^	
	Crude OR/ *β* (95% CI) ^a^	Adjusted OR/ *β* (95% CI) ^a,c^	Crude OR/ *β* (95% CI) ^a^	Adjusted OR/ *β* (95% CI) ^a,c^	Crude OR/ *β* (95% CI) ^a^	Adjusted OR/ *β* (95% CI) ^a,c^
Degree of Intelligence	1.03 (0.79~1.32)	1.22 (0.72–1.89)	0.93 (0.72~1.20)	1.08 (0.84–1.77)	0.69 (0.53~0.90) *	0.77 (0.49–0.95) *
IQ	0.18 (−3.55~3.91)	0.33 (−3.71~4.36)	−1.97 (−5.70~1.76)	−1.32 (−5.48~2.84)	−5.71 (−9.44~−1.98) *	−5.85 (−10.13~−1.57) *
5-HT	−36.79 (−77.65~4.07)	−27.41 (−71.36~16.55)	−38.28 (−80.99~4.43)	−17.13 (−63.75~29.49)	−69.24 (−114.85~−23.64) *	−55.61 (−104.67~−6.54) *
NE	−238.68 (−424.14~−53.22) *	−182.38 (−374.53~9.77)	−102.67 (−300.37~95.03)	−11.02 (−216.24~194.20)	−127.36 (−333.21~78.49)	−76.57 (−288.67~135.53)
DA	2.84 (0.50~5.17) *	3.34 (0.78~5.89) *	2.24 (−0.15~4.64)	3.08 (0.42~5.75) *	−0.42 (−3.01~2.18)	−0.03 (−2.85~2.80)
E	61.64 (−47.66~170.94)	75.41 (−41.72~192.54)	37.56 (−78.92~154.03)	37.36 (−90.11~164.83)	94.92 (−21.56~211.39)	115.00 (−10.50~240.50)

* *p* < 0.05; ^a^. OR: degree of intelligence; *β:* IQ, 5-HT, NE, DA, E; ^b^. N-L group served as the control group; ^c^. Mixed factors of adjustment: the confounding factors of degree of intelligence and IQ adjusted Stress, Often catching a cold, Age, and BMI. The confounding factors of 5-HT, NE, DA, and E adjusted FT3, FT4, and TSH; ^d^. WI-WF group: the subjects were grouped according to different concentrations of iodine and fluoride in water; see Section 2.1 for details.

**Table 4 nutrients-16-02913-t004:** The effect of urinary iodine and urinary fluoride on thyroid health in schoolchildren.

Factor	UI-UF Group ^d^					
	L-H ^b^		H-L ^b^		H-H ^b^	
	Crude OR/ *β* (95% CI) ^a^	Adjusted OR/ *β* (95% CI) ^a,c^	Crude OR/ *β* (95% CI) ^a^	Adjusted OR/ *β* (95% CI) ^a,c^	Crude OR/ *β* (95% CI) ^a^	Adjusted OR/ *β* (95% CI) ^a,c^
Thyroid nodule	0.00 (0.00-Inf)	0.00 (0.00-Inf)	0.45 (0.11–1.73)	0.47 (0.12–1.86)	3.19 (1.38–7.36) *	2.85 (1.22–6.69) *
Thyroid abnormal	0.00 (0.00-Inf)	0.00 (0.00-Inf)	1.10 (0.50–2.38)	1.07 (0.48–2.41)	3.30 (1.75–6.24) *	2.99 (1.56–5.71) *
Thyroid disease	1.13 (0.44–2.92)	1.08 (0.41–2.81)	0.91 (0.50–1.64)	0.97 (0.53–1.78)	1.01 (0.59–1.72)	1.10 (0.90–1.34)
Tvol	−0.10 (−0.70~0.51)	−0.17 (−0.77~0.44)	0.38 (0.01~0.74) *	0.39 (0.02~0.76) *	1.59 (1.26~1.92) *	1.55 (1.21~1.89) *
FT3	0.12 (−0.20~0.44)	0.10 (−0.22~0.43)	−0.01 (−0.20~0.19)	−0.00 (−0.20~0.20)	−0.36 (−0.54~−0.18) *	−0.35 (−0.53~−0.16) *
FT4	0.62 (−0.27~1.50)	0.76 (−0.13~1.65)	−0.31 (−0.85~0.23)	−0.33 (−0.88~0.21)	0.37 (−0.13~0.87)	0.43 (−0.08~0.94)
TSH	−0.00 (−0.75~0.74)	−0.03 (−0.79~0.72)	0.34 (−0.13~0.81)	0.28 (−0.20~0.76)	0.31 (−0.11~0.74)	0.27 (−0.16~0.71)

* *p* < 0.05; ^a^. OR: thyroid nodule, goiter, thyroid abnormal, thyroid disease; *β*: Tvol, FT3, FT4, TSH; ^b^. L-L group as the control group; ^c^. Adjustment confounding factors: the confounding factors of thyroid nodule, goiter, thyroid normal, thyroid disease, Tvol, FT3, FT4, and TSH adjusted Kelp weeded Soup, Salt type, and Age; ^d^. UI-UF group: the subjects were divided into groups according to different urine iodine and urine fluoride concentrations; see Section 2.1 for details.

**Table 5 nutrients-16-02913-t005:** The effect of urinary iodine and urinary fluoride on the intelligence of schoolchildren.

Factor	UI-UF Group ^d^					
	L-H ^b^		H-L ^b^		H-H ^b^	
	Crude OR/*β* (95% CI) ^a^	Adjusted OR/*β* (95% CI) ^a,c^	Crude OR/*β* (95% CI) ^a^	Adjusted OR/*β* (95% CI) ^a,c^	Crude OR/*β* (95% CI) ^a^	Adjusted OR/*β* (95% CI) ^a,c^
Degree of Intelligence	0.53 (0.11–2.48)	0.47 (0.10–2.24)	1.37 (0.67–2.79)	1.32 (0.64–2.72)	1.46 (0.76–2.79)	1.27 (0.65–2.46)
IQ	−6.72 (−12.87~−0.58) *	−4.67 (−10.46~1.12)	−2.40 (−6.09~1.28)	−3.14 (−6.66~0.38)	−3.68 (−7.04~−0.33) *	−2.47 (−5.68~0.74)
5-HT	0.46 (−67.72~68.63)	−16.98 (−83.95~49.99)	45.24 (2.69~87.78) *	36.51 (−7.33~80.35)	9.87 (−28.90~48.63)	2.98 (−37.95~43.92)
NE	77.92 (−246.77~402.62)	−1.82 (−312.72~309.08)	−10.71 (−208.51~187.10)	−67.51 (−266.17~131.14)	86.62 (−91.65~264.88)	37.55 (−145.86~220.96)
DA	1.41 (−2.16~4.97)	0.64 (−2.96~4.25)	1.70 (−0.53~3.94)	1.47 (−0.88~3.82)	−1.03 (−3.05~0.98)	−1.45 (−3.64~0.73)
E	24.84 (−191.14~240.82)	18.10 (−201.00~237.19)	−60.39 (−173.61~52.83)	−94.76 (−214.31~24.80)	11.43 (−90.57~113.44)	0.04 (−110.52~110.60)

* *p* < 0.05; ^a^. OR: degree of intelligence; *β:* IQ, 5-HT, NE, DA, E; ^b^. L-L group as the control group; ^c^. The confounding factors of adjustment: the confounding factors of degree of intelligence and IQ adjusted the precedence conditions and Age. The confounding factors of 5-HT, NE, DA, and E adjusted FT3, FT4, and TSH; ^d^. UI-UF group: the subjects were divided into groups according to different urine iodine and urine fluoride concentrations; see Section 2.1 for details.

**Table 6 nutrients-16-02913-t006:** Interaction between iodine and fluorine on thyroid and intelligence of school-age children.

Index	Thyroid Nodule			IQ		
	UI	UF	UI & UF	UI	UF	UI & UF
Coefficients (B ± SE)	−14.97 ± 843.46	−0.81 ± 0.69	1.16 ± 0.43	−0.02 ± 0.01	−3.02 ± 1.44	0.01 ± 0.00
OR (95% CI)	0.00 (0.00-Inf)	0.45 (0.11–1.73)	3.19 (1.38–7.36)	0.98 (0.97–1.00)	0.05 (0.00–0.83)	1.01 (1.00–1.01)
*p*	0.99	0.24	0.01 **	0.02 **	0.04 **	0.07 *
RERI (95% CI)	1.37 (0.74~2.01)			0.97 (0.82~1.13)		
AP (95% CI)	3.08 (0.32~5.83)			0.97 (0.82~1.11)		

** *p* < 0.05; * *p* < 0.1.

## Data Availability

The original contributions presented in this study are included in the article and Appendix A, further inquiries can be directed to the corresponding authors.

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
