# Peer review of "The Impact of Exposure to Iodine and Fluorine in Drinking Water on Thyroid Health and Intelligence in School-Age Children: A Cross-Sectional Investigation"

_nutrients, 2024, doi:10.3390/nu16172913_

Round 1
Reviewer 1 Report
Comments and Suggestions for Authors
Dear Authors,
The article presented to me for evaluation is very interesting and a great contribution to science. The topic is timely and related to the risks affecting the health of children. A great advantage of the experiment is a large study group which increases the value of the publication.
As a reviewer, I have no substantive comments and recommend the article for publication after minor editorial corrections.
1. literature and citations should be adjusted to the requirements of the journal
2. chapters should be properly numbered.
Page 13 is 2.6 - should be 3.6
similarly 2.7 - should be 3.7
However, these are minor comments that do not affect the final evaluation of the article
Author Response
Comments 1: Literature and citations should be adjusted to the requirements of the journal.
Response 1: Thank you for pointing this out. We agree with this comment. Therefore, we refer to the requirements of the MDPI official website regarding the format of references and make corresponding modifications in the "References" section of the manuscript. Please see the revised manuscript's lines 359-441 on pages 19-21.
Comments 2: Chapters should be properly numbered. Page 13 is 2.6 - should be 3.6; similarly 2.7 - should be 3.7.
Response 2: Thank you for pointing out the above problems. The article number has been revised and checked in the manuscript. See pages 14-15 for details.

Reviewer 2 Report
Comments and Suggestions for Authors
In this study, the authors evaluated the association of the combined exposure to different levels of iodine and fluoride in drinking water with thyroid health and intellectual status in schoolchildren. The study is interesting and provides a huge amount of results. The study design is appropriate, and the results are overall clear and widely discussed. I only have minor comments.
Abstract.
Please provide the full names of abbreviations used.
Introduction
The authors use “iodine” and “iodide” alternately. The correct term is “iodine”.
Methods
Lines 145,147. The term “obvious” should be replaced with “overt”.
Results
Line 215. Please provide the full name of “UI” and “UF”.
Tables. What are the WI-WF group and the UI-UF group? Please insert a list of abbreviations below the tables.
Page 8, line 19. Please replace “adjustment” with “adjusted”.
Page 8, lines 25-26. I cannot find these values in tables s2 and 3.
Table 3. What does E stand for?
Page 9, line 38. The adjusted OR reported in table 4 is 1.55 (1.21~1.89).
Page 9, lines 41, 44. Please compare the ORs reported in the text with those in Table 4.
Discussion
Page 16, line 221. Please specify the “other areas”.
Page 18, line 291. Please provide here the full name of HPG and not on line 306.
Comments on the Quality of English LanguageMinor English editing
Author Response
Comments 1: Abstract: Please provide the full names of abbreviations used.
Response 1: Thank you very much for pointing this out. We have checked and revised the abbreviations in the abstract and added the full names to the abbreviations that first appeared. In the abstract in the received manuscript, "IQ" was changed to the full name "intelligence quotient (IQ)". "FT3", "FT4", and “TSH” have been changed to the full name "triiodothyronine (FT3)", "free tetraiodothyronine (FT4)", and “thyroid stimulating hormone (TSH)”. In addition, “DA”, “NE”, and “5-HT” have been changed to “dopamine (DA)”, “noradrenaline (NE)”, and “5-hydroxytryptamine (5-HT)”.
Comments 2: Introduction: The authors use “iodine” and “iodide” alternately. The correct term is “iodine”.
Response 2: Thank you for your suggestion, which can improve the accuracy and professionalism of the article. We have replaced the word "iodide" with “iodine” in the article. Such as lines 14, 18, 46, 58, 61, 74(before Results 3.2), 139, 189, 191, 212, 222, 232, 233, 235, 249, 250, 282, 312, 322, 327, 329 and 335.
Comments 3: Methods: Lines 145,147. The term “obvious” should be replaced with “overt”.
Response 3: Thank you for your suggestion to make the language expression of the article more accurate. We have made corresponding revisions. For details, in lines 151 and 153 of the revised manuscript, "obvious hypyroidism" has been changed to "overt hypyroidism".
Comments 4: Results: Line 215. Please provide the full name of “UI” and “UF”.
Page 8, line 19. Please replace “adjustment” with “adjusted”.
Response 4: Thank you for your suggestion. After inspection, "UI" and "UF" first appeared in the third paragraph of "Introduction", "And the correlation between TSH and uric fluoride (UF) will be influenced by uric iodine (UI)", and will not be supplemented in "Results". "adjustment β" in the third line of "Results 3.3" was replaced by "adjusted β". Thank you for pointing out this error.
Comments 5: Tables: What are the WI-WF group and the UI-UF group? Please insert a list of abbreviations below the tables.
Table 3. What does E stand for?
Page 8, lines 25-26. I cannot find these values in tables s2 and 3.
Page 9, line 38. The adjusted OR reported in table 4 is 1.55 (1.21~1.89).
Page 9, lines 41, 44. Please compare the ORs reported in the text with those in Table 4.
Response 5: Thank you for your suggestion. We have made the following corresponding changes:
- Notes about the WI-WF group and the UI-UF group have been added under Table 1-5, for example, Table 1. has added the note“ WI-WF group: the subjects were grouped according to different concentrations of iodine and fluoride in water; UI-UF group: the subjects were divided into groups according to different urine iodine and urine fluoride concentrations; see ‘2.1. Study population and sampling’ for details.”; Table 2-3 has added the note “d. WI-WF group: the subjects were grouped according to different concentrations of iodine and fluoride in water; see ‘2.1. Study population and sampling’ for details.”; Table 4-5 has added the note “d. UI-UF group: the subjects were divided into groups according to different urine iodine and urine fluoride concentrations; see ‘2.1. Study population and sampling’ for details.”;
- E in Table 3 stands for norepinephrine, and this abbreviation first appeared in the third paragraph of "Materials and methods 2.2", “The contents of norepinephrine (NE), epinephrine (E), dopamine (DA), and 5-hydroxytryptamine (5-HT) were detected by an ELISA kit built in Nanjing.”;
- The original text of Page 8, lines 25-26 is“Compared with N-L group, the adjusted β of IQ in H-H group is -5.85 (95%CI: -10.13, -1.57) (see Table 3), and the IQ score in H-H group is significantly different from other groups (adjusted p<0.05) (see Table s2).”The tables and specific values mentioned in the original text have been marked with red boxes as follows (See PDF file), and Table s2 is located in the "Supplementary Data" file.
- Thank you very much for your suggestions for revision! Some data in "Results 3.4" are different from Table 4, and have been modified and checked. For example, the third line was changed to "1.55 (95%CI: 1.21, 1.89)" after checking, and the sixth and seventh lines were changed to "2.85 (95% CI: 1.22, 6.69) and 2.99 (95% CI: 1.56, 5.71)" after checking. Besides, we checked the data in other results and modified "(see Table 3)" in the last line of the first paragraph of "Results 3.3".
Comments 6: Discussion: Page 16, line 221. Please specify the “other areas”.
Page 18, line 291. Please provide here the full name of HPG and not on line 306.
Response 6: Thank you for your suggestion. We have made the following corresponding changes:
- Thank you for your suggestion to make the article more accurate. We have changed "other areas" to "non-high iodine or non-high fluorine areas".
- In line 308 of the revised manuscript, we added the full name of the HPG axis.
Point 1: Minor English editing
Response 1: Thank you for your suggestion. We have invited a native English speaker to help to revise the manuscript. We made a lot of changes, such as the "hyperparathyroidism" in the revised manuscript line 153 has been changed to "hyperthyroidism"; the keyword “fluorine” has changed to "fluoride".

Reviewer 3 Report
Comments and Suggestions for Authors
Thank you very much for allowing me to review this paper. The topic is very interesting, and this article will contribute to increase the knowledge in this field. The combined effect of iodine and fluoride exposure in drinking water on thyroid function has been evaluated on 399 Chinese school-aged children, grouped according to the elements’ concentration in drinking water. The investigated variables were thyroid hormone levels, the gland volume and the presence of nodules, IQ, and neurotransmitter levels (the latter were used to explore the potential relationship between thyroid function and intelligence). Iodine and fluoride (in water and urine) resulted positively correlated with thyroid volume and the incidence of thyroid nodules, and negatively with IQ. The effect resulted to be synergic, affecting thyroid hormone secretion and -by a feed-back mechanism- TSH levels, so influencing QI (FT3 and TSH negatively and FT4 positively). A supplemental mechanism on QI can be mediated by the reduction of the positive effects on neurotransmitter secretion elicited by thyroid hormones.
Despite the study is rich of data, very well performed and described, some amendments are warranted:
-Many typos in the text should be corrected, and the reference style is not appropriate for the journal
-In the abstract, it should be added that the study was conducted in China
-Introduction: it is clear and rich of information
-Material and methods are thoroughly described, in line 147 “hyperparathyroidism” should be corrected in “hyperthyroidism
-Results: lines 170-172, 211-215: the sentences are not clear, please rephrase them. This paragraph contains in the text data already presented in the tables. Please consider removing repetitions of numbers in the text, to make fluid the reading.
-Discussion: it repeats some concepts and hypotheses already anticipated in the previous parts: please select which collocation is more suitable.
Author Response
Comments 1: Many typos in the text should be corrected, and the reference style is not appropriate for the journal.
Response 1: Thank you for your correction. The citation format has been revised according to the requirements of the magazine. See the "References" section of the manuscript for details. The following amendments have been made to the inaccuracies or irregularities in this article:
â‘ We unify the existing "school-aged children" and "school-age children" in the original manuscript into "school-age children";
â‘¡The note of Table 6 was changed to "p<0.05; *p<0.1.”;
â‘¢In line 36, we amend the keyword to "fluoride".
Comments 2: In the abstract, it should be added that the study was conducted in China.
Response 2: I quite agree with your suggestion, and "This study investigated 399 school-age children in Tianjin, China, collected drinking water samples from areas where school-age children lived, and grouped the respondents according to iodine and fluoride levels." is shown in line 16 of the revised manuscript.
Comments 3: Material and methods are thoroughly described, in line 147 “hyperparathyroidism” should be corrected in “hyperthyroidism”.
Response 3: Thank you for your suggestions to make the article more accurate and professional. The "hyperparathyroidism" in the revised manuscript line 153 has been changed to "hyperthyroidism".
Comments 4: Results: lines 170-172, 211-215: the sentences are not clear, please rephrase them. This paragraph contains in the text data already presented in the tables. Please consider removing repetitions of numbers in the text, to make fluid the reading.
Response 4:
- Thank you for your suggestion. Please allow me to explain the interpretation of the title of Fig. 5 in lines 170-172 of the original manuscript, which may cause you to have difficulty reading it due to the proximity of the expressions. Lines 170-172 in "Results" are the annotations of Figure 5, in which Figure A mainly shows the path analysis results of the potential relationship between urinary iodine and urinary fluoride of school-age children and their intellectual health through monoamine neurotransmitters and thyroid hormones, and Figure B is the summary of the path analysis of Figure A; Figure C is the path analysis of the potential relationship between urinary iodine and urinary fluoride in school-age children through thyroid hormones and thyroid health, and Figure D is the summary of the path analysis of Figure C.
- Lines 211-215 in "Discussion" have been changed to “There may be some antagonism between water iodine and fluoride in the thyroid, and the influence of fluoride on thyroid may reduce the influence of iodine. In the case of excessive iodine in water, the risk of thyroid nodules and thyroid abnormalities decreased by 0.12 and 0.33 times for every unit of fluoride in water, while Tvol, FT4, and TSH decreased by 0.31mL, 1.08pmol/L, and 0.66μIU/mL on average, but the effect was weak, which may affect the activity of TSH by inhibiting the activity of thyroidic acid cyclase (Jiang and Guo et al., 2016; Jenq and Jap et al., 1993).”, see lines 218-224.
Comments 5: Discussion: it repeats some concepts and hypotheses already anticipated in the previous parts: please select which collocation is more suitable.
Response 5: Thank you for your suggestion. We made some appropriate modifications and adjustments to the discussion part, such as the sentence “Due to the topography, there is a significant correlation between iodine and fluoride content in groundwater in North and East China of China (Huang and Guo et al., 2023).” has been deleted; modifying lines 261-264 to "By fitting the nonlinear relationship between urine iodine, urine fluoride, and intelligence, it is found that iodine, fluoride, and intelligence evaluation scores of school-age children are indeed negatively correlated as previously assumed. ". Lines 278-279 added, "In this study, the role of monoamine neurotransmitters in the above-mentioned influencing pathways was discussed.”
